# Compact UHF Circularly Polarized Multi-Band Quadrifilar Antenna for CubeSat

**DOI:** 10.3390/s23125361

**Published:** 2023-06-06

**Authors:** Manh Thao Nguyen, Fabien Ferrero, Le-Huy Trinh

**Affiliations:** 1Universite Cote d’Azur, Laboratoire d’Electronique, Antennes et Telecommunications (LEAT), 06903 Sophia Antipolis, France; 2Centre National de Recherche Scientifique (CNRS), UMR7248, 06903 Sophia Antipolis, France; 3Faculty of Computer Engineering, University of Information Technology, VNU-HCM, Ho Chi Minh City 700000, Vietnam; huytl@uit.edu.vn

**Keywords:** circular polarization, CubeSat, multi-band antenna, antenna on structure

## Abstract

This article presents a multi-band right-hand circularly polarized antenna designed for the Cube Satellite (CubeSat). Based on a quadrifilar structure, the antenna provides circular polarization radiation suitable for satellite communication. Moreover, the antenna is designed and fabricated using two 1.6 mm thickness FR4-Epoxy boards connected by metal pins. In order to improve the robustness, a ceramic spacer is placed in the centerboard, and four screws are added at the corners to fix the antenna to the CubeSat structure. These additional parts reduce antenna damage caused by vibrations in the launch vehicle lift-off stage. The proposal has a dimension of 77 × 77 × 10 mm^3^ and covers the LoRa frequency bands at 868 MHz, 915 MHz, and 923 MHz. According to the measurements in the anechoic chamber, antenna gains with the values of 2.3 dBic and 1.1 dBic are obtained for the 870 MHz and 920 MHz, respectively. Finally, the antenna is integrated into a 3U CubeSat that was launched by a Soyuz launch vehicle in September 2020. The terrestrial-to-space communication link was measured, and the antenna performance was confirmed in a real-life scenario.

## 1. Introduction

The development of the new space industry is disrupting the space market by making satellite fabrication and launch more affordable. Especially miniature satellites, such as the so-called CubeSat [1], enable more accessible access to space. Most of these satellites have dimensions from 1U to 12U and are deployed in a low Earth orbit (LEO) (with U considered as 1 unit with a size of 10 × 10 × 10 cm^3^). Typical applications of this system are mentioned, such as communication, Earth observation, and space exploration [2,3,4]. In the Earth-to-space communication application, thanks to integrating different wireless communication technologies, satellites can connect to terrestrial Internet of Things (IoT) devices in various areas, from the metropolitan zone to the remote zone.

In the recent decade, several highly sensitive modulations known as low-power wide-area networks (LP-WAN) have been proposed to enable connectivity from small terminals to space [5,6]. With long-range connectivity higher than 1000 km, LEO satellites with LP-WAN transceivers based on LoRa ensure the ability to communicate with terrestrial systems [7]. Several companies have successfully implemented satellite systems based on LoRa modulation such as Lacuna Space, Swarm technologies, and Fossa Systems [8,9,10]. It confirms that the feasibility will be expanded shortly [11].

A CubeSat includes a communication subsystem for satellite–ground links. The quality of connectivity strongly depends on the antenna system’s performance integrated inside the satellite. Antenna design is a challenge for researchers due to the limitations of size, mass, operating frequency, and required bandwidth. In [12,13,14,15,16], various antenna types proposed for CubeSat were designed and analyzed, such as monopole, dipole, helical, patch, slot, phase array, reflect array, and metamaterial-based antennas.

Regarding operating frequency, antennas operating in a band higher than 1 GHz are often preferred due to their small size and ability to support broadband communications. A wideband patch antenna has been reported that can cover 1.6 to 2.7 GHz with a maximum gain of 8.5 dBi [17]. Due to the linear polarization (LP), the misalignment between the CubeSat and terrestrial station could affect the quality of communication. A circularly polarized (CP) antenna is proposed to solve this problem. In [18], a loaded slot antenna is introduced with a size of 100 × 100 × 1.6 mm^3^ and an axial ratio bandwidth of 18.75%. An antenna is designed on low-loss Rogers substrates and provides an omnidirectional radiation pattern with a maximum gain of 8 dBic.

On the other hand, directional antennas have also been extensively studied [19,20,21,22,23,24,25,26]. The article [19] presents a mechanical structure with a horn antenna and reflect array. Due to the 900 elements of coupled loops, the proposal provides a high-gain CP antenna for the K-Band frequency. Some studies suggest using fixed antennas to reduce the risk of deploying mechanical systems in space. The articles [20,21,22] propose using feeding networks to distribute signals to radiating elements with suitable phases and then to generate the directional CP pattern. Furthermore, metamaterials are also mentioned in [25,26].

Differently from the previous examples, sub-GHz bands enable longer transmission distances thanks to the lower path loss and easier obstacle penetration. In article [27], an omnidirectional LP antenna was reported at an operating frequency of 920 MHz. The meander line structure significantly reduced the antenna size with a surface of 50 × 80 mm^2^ and realized a gain of −1.8 dB. Another study proposed a slot-based antenna that supports the MIMO scheme mentioned in [28]. Due to the folding of the slot shape, the antenna’s resonant frequency band is reduced to 430–510 MHz while maintaining the dimensions within 100 × 100 mm^2^. However, this antenna requires a large clearance area, so mounting it on satellites will be pretty tricky. To solve this problem, by analyzing the characteristic mode of the satellite structure, the paper [29] presents a no-need-to deploy antenna that operates at 433 MHz within a 1U dimension. However, the bandwidth is quite narrow, about 3 MHz, which could limit the application of this antenna. In addition, the linear polarization properties and the directional radiation pattern could reduce the quality of the channel and the communication distance. In [30], a proposal based on a meandered folded-shorted patch array is reported. By using a complex coupler network, the input signal could be divided into four paths with the same amplitude. Furthermore, due to the phase shift of 90° between each port, a circular polarization feature is achieved. However, because of the narrow impedance bandwidth, the high-cost substrate material, and fragile antenna structure, this proposal could not be suitable for satellite application.

Designing the antenna with a compact size to be compatible with CubeSat integration and operation at low frequencies with specific properties in polarization and pattern is a considerable challenge. In [31], two frequency reconfigurable antenna structures have been proposed. This system consists of a directional CP antenna based on the quadrifilar helix form and a directional LP antenna based on the Yagi-Uda structure. These antennas support the tunable frequency scheme to extend the effective bandwidth thanks to a gearing system. Recently, several miniature circularly polarized antennas based on a multi-element approach have been proposed to combine a compact form factor with a wide-beamwidth circular polarization [32,33,34].

Lacuna Space is a satellite IoT network provider founded in 2017 [35]. Lacuna Space offers a low Earth orbit satellite network using sub-GHz unlicensed bands. The network is natively compatible with the terrestrial LoRaWan network using the same frequency bands. Terrestrial terminals need to know the Lacuna Space constellation almanac in order to wake up at the right time to transmit to the CubeSat. The constellation almanac contains the Keplerian element of each CubeSat orbit, which enables the exact position of any CubeSat to be computed for a given time. Due to atmospheric drag, and moon and sun influence, the almanac is valid for up to one month; then, it needs to be updated regularly. The CubeSat described in this paper aims to broadcast from space the almanac information on the Lacuna Space constellation in the 870 MHz and 920 MHz bands to terrestrial nodes. Circular polarization is required to avoid misalignment between the transmitter and receiver and to reduce the Faraday effect. Additionally, the volume available for the antenna system is extremely limited as the antenna is positioned on the 1U face. Considering the mechanical frame structure and depending on the launching system, the maximal height is 10 mm. A non-deployable solution is chosen to limit any risk linked to the deployment system.

This article presents a low-profile, miniature, and multi-band antenna for 3U CubeSat that supports terrestrial ISM bands with right-hand circular polarization. The antenna is designed based on the combination of a broadband power splitter structure and inverted-F antenna elements (IFA) to obtain the directional radiation pattern. In addition, the proposed design provides bandwidth expansion thanks to coupling with a shorted parasitic element. A CubeSat-suitable robust structure is archived using the plastic bumper as the spacing. The antenna is manufactured, mounted on the satellite frame, and then measured in the chamber to validate the concept. Finally, the antenna was integrated within LS3 Lacuna Space CubeSat, launched in September 2020, then commissioned, and used for in-flight experiments in February 2021.

## 2. Antenna System Design

As mentioned above, the satellites supporting UHF communication need to cover multiple bands for different geographical areas. When the satellite reaches different areas, it will adjust its transmission and reception frequencies to the bands allowed in those regions, such as 868 MHZ in Europe, 915 MHz in the Americas, and 915 MHz and 923 MHz in Asia. Therefore, antennas used for satellites need to have multi-frequency band features to ensure worldwide continuous operation. In addition, to optimize transmission quality and to reduce the interference caused by ground-based systems, a high gain with circular polarization and a wide beamwidth radiation pattern are required. For these reasons, Lacuna Space decided to use right-hand circular polarization (RHCP) for uplink and downlink communications.

To integrate the radiation structure on the 1U face of the 3U CubeSat, the maximal dimensions of the structure were 77 × 77 mm^2^ and 10 mm in height. The radio frequency connector was placed on the bottom layer of the bottom substrate. Based on the literature, a single resonance quadrifilar structure with such a form factor limitation would reach a 30 MHz frequency bandwidth. In order to extend the frequency band operation, a reconfigurable solution based on digital tunable capacitors or pin diodes was considered too risky. In order to provide a passive dual-band antenna, parasitic elements have been extensively used with an inverted F antenna [32]. Then, to meet the project requirements, we proposed to combine a quadrifilar structure with parasitic elements. Similar to [32], two thin printed circuit board (PCB) substrates were used. The feeding circuit and connector were placed on the bottom PCB, and antenna traces were placed on the top PCB. The two substrates were connected using 10mm long vertical wires soldered to both PCBs. To provide good mechanical characteristics and a reduced production time, 1.6 mm thick FR-4 substrate material ($\epsilon_{r}$ = 4.4 and tan δ = 0.02) was chosen to manufacture the two substrates. The use of FR-4 material is not optimal to maximize the radio frequency performance of the antenna but it is qualified for space operation and ensures good mechanical properties to handle the vibration at the satellite launching phase. In addition, a 25 mm diameter ceramic ring was placed in the center of the structure to improve mechanical robustness.

### 2.1. Antenna Elements

Inverted F antenna radiating elements are designed using the bottom substrate as a ground plane. The dual-band characteristic is obtained by adding a parasitic element placed in the inner part of the structure and connected to the ground. Several configurations of the driven and parasitic elements were investigated to find out the best topology to maximize frequency bandwidth and gain. Parametric simulations show that the driven element must be placed on the edge of the structure, with the driven element placed at 8mm in the center of the structure. The parasitic element shorting pin is aligned with the feeding element shorting pin. The width and length of the driven and parasitic elements are carefully optimized to maximize the performance in both bands.

As shown in Figure 1a, the driven elements are placed on the edge of the ground plane with an optimized length to provide resonance at 868 MHz. The parasitic element is placed inside the structure, and the length is shorter and optimized to operate at 920 MHz. The driven element input impedance is tuned to 50 Ω by adjusting the distance between the feed and the shorting pins.

Four 8 mm diameter holes are inserted into the top substrate to enable the screwing of the antenna structure to the 3U frame. The M3 metallic screws are also used for grounding purposes on the CubeSat metallic structure. The 3D and top view layouts are shown in Figure 1b.

Surface currents at 870 MHz and 920 MHz are presented in Figure 2. At 870 MHz, surface currents are strong on the driven elements with low intensity on the parasitic elements. At 920 MHz, the driven elements excite the parasitic element and strong current intensity is visible on both branches.

### 2.2. Feeding Circuit

As described in the previous section, the radiating structure is composed of four independent IFAs with sequential rotation. To radiate right-hand circular polarization, each element must be fed equally with a sequential phase shift of 0°, 90°, 180°, and 270°. Various solutions have been proposed to create such a feeding mechanism in the literature. For example, three hybrid branch-line couplers and a 90° phase shifter are combined, which is presented in [36]. This solution provides a low insertion loss and good balance but requires the integration of three branchline couplers and delay lines. This structure would be difficult to integrate with the limited surface available (due to the ceramic ring in the center).

In order to miniaturize the circuit, the proposed feeding network consists of a balun and two hybrid couplers, as shown in Figure 3, which is inspired by the Marchand structure in [37], the balun is designed using the microstrip line with a phase shift of 180° between the two output ports. The center microstrip quarter wave transmission line is terminated with a short circuit. The upper and lower microstrip lines are coupled with the center line thanks to 0.3 mm width slots. The balun output ports are connected to the two hybrid couplers to provide four equal signals with the phase shifts of 0°, 90°, 180°, and 270°. To avoid unwanted reflection effects due to the radiating elements mismatching, the coupler isolation ports are matched to 50 Ω using discrete resistors. These ports are mentioned as Port 2 and Port 3 in the block diagram of the feeding network structure (Figure 3).

The proposed design is illustrated in Figure 4a, with the yellow element being the broadband 180° balun and the orange parts being the hybrid couplers. In order to minimize the dimension of the feeding circuit, meander microstrip lines are used. In addition, the blue elements are presented as 50 Ω transmission lines that will be connected to the radiating element on the top PCB. The main RF connector (Port 1) is located on the bottom part of the substrate and is directly connected to the unbalanced input of the balun.

### 2.3. Electromagnetic Modeling of the Proposed Structure

Ansys HFSS software is used to simulate the proposed antenna system; then, the design concept could be confirmed. In the next section, two different cases are presented in order to understand the effect of the 3U satellite structure on antenna performance. The case of “without structure” refers to the antenna alone, as shown in Figure 1. Then, the case of “with structure” refers to the antenna mounted on the 3U CubeSat metal frame. In this case, the satellite structure is added and analyzed to verify its influence on the antenna features. As shown in Figure 4b, a simplified CubeSat structure with an external conductive frame is considered. It can be seen that four frame corners extend 10 mm around the antenna, which could affect the impedance matching and the radiation pattern of the proposed antenna. It should be noted that the maximal dimension of the 3U satellite is 34 cm, which is exactly the wavelength at 870 MHz. The simulated results are shown in Figure 5 without and with a satellite structure. The reflection coefficient on port 1 and the transmitted power to ports 2 and 3 (matched isolated port) are considered in order to analyze the global performance of the structure. The most important parameter is $S_{11}$ in order to provide an optimal input impedance to the power amplifier. The transmission to ports 2 and 3 should be minimized to maximize the antenna’s total efficiency. A −10 dB criterion is targeted for $S_{11}$, $S_{21}$, and $S_{31}$ on the entire frequency band ranging from 860 to 930 MHz. According to Figure 5a, the $S_{11}$ curves of these cases are quite different. These dissimilarities could be explained by the influence of a large metallic part placed under the antenna. However, they are shown to respect the S-parameter criteria for the required frequency band.

The gain achieved on the broad side versus the frequency is presented in Figure 5b. Without and with the CubeSat structure, the right-hand circular polarization is achieved with an axial ratio lower than 3 dB (a 3 dB axial ratio is equivalent to a 15.35 dB difference between the RHCP and LHCP gains [38]). Without the structure, the simulated realized gain is higher than 1.6 dBic on the 860–930 MHz band. However, when the antenna is mounted on the CubeSat structure, the simulated realized gain is lower, which is only more than 0 dBic in the whole band. Several reasons can explain this effect. First, the reflection coefficient of the antenna-only case is better than the complete system. Second, the total directivity with the structure is slightly lower because of vertical currents flowing on the CubeSat chassis and increasing the antenna beamwidth.

Without the CubeSat structure, the realized gain radiation patterns at 870 MHz and 920 MHz are presented in Figure 6. A 120° beamwidth of RHCP is obtained in both bands, with 2.5 dBic as gain. Regarding the LHCP radiation pattern, the proposed antenna obtained a gain of −2.5 dBic and is radiated in the opposite direction. When the antenna system is mounted on the metal satellite frame, Figure 7 shows a wider beamwidth radiation pattern. However, a lower RHCP gain is obtained, and back LHCP radiation is also reduced on both bands. The circular polarization feature is also reported in Figure 8. According to these results, the proposal has a 3 dB AR beamwidth of 120° toward the z-axis. Moreover, due to the effect of the satellite structure, the obtained axial ratio is more efficient than the antenna-only case.

## 3. Prototype Measurement Results

The proposed concept is manufactured and assembled using two FR4 dual-layer substrates ($\epsilon_{r}$ = 4.4 and tan δ = 0.02). A ceramic ring is used to fix the structure. From EM simulation, it was possible to predict that the ceramic ring will not have any impact on the radiofrequency performance of the structure. Twelve 0.8 mm diameter vertical pins are soldered to connect the two substrates. This structure ensures solid mechanical properties to reduce any risk of vibration during launch vehicle takeoff. Four holes are etched into the top substrate to screw the antenna on the CubeSat structure. In Figure 9, the top and side views of the prototype are shown. The radio frequency performance of the proposed design is measured without and with the 3U structure composed of a 10 × 10 × 30 mm aluminum frame in the experiments.

The measured reflection coefficients of the antenna without and with the structure are presented in Figure 10a. An $S_{11}$ lower than −10 dB is obtained in the 860–930 MHz band, which shows good agreement with the simulated results. The radiating characteristics are performed in an anechoic chamber. The measured broadside gain versus frequency is shown in Figure 10b. A right-hand circular polarization with an axial ratio lower than 3 dB is obtained from 860 MHz to 930 MHz. As anticipated from the 3D EM modeling, the broadside gain in the prototype mounted on the CubeSat structure is relatively lower than that in the antenna-only case.

For the antenna alone, the radiation patterns of the real gain at 870 MHz and 920 MHz are shown in Figure 11. According to this report, a good agreement is obtained with the simulated results. An RHCP 122° beamwidth pattern is measured with peak gains of 2.3 dBic and 1.1 dBic at 870 MHz and 920 MHz, respectively. In the case where the proposed antenna is integrated with the structure, the realized gain radiation patterns at 870 MHz and 920 MHz are shown in Figure 12. Similarly to the simulation results, the gain is decreased due to the contribution of the vertical currents flowing on the 3U structure. The 3 dB beamwidth is wider, with 132° and 129° at 870 MHz and 920 MHz, respectively. The axial ratio level versus theta for different configurations is also computed and shown in Figure 13. An axial ratio level lower than 3 dB can be observed over a 120° aperture beamwidth for both configurations. Some asymmetry in the axial ratio results can be observed.

These measurement results validate the performance of the proposed structure for the targeted application.

## 4. In-Flight Antenna Performance

A picture of the antenna integrated within the 3U structure is shown in Figure 14. Solar panels are deployed, and the proposed UHF antenna is visible on the front of the satellite. The 3U CubeSat, named LS3, was launched on 28 September 2020 by the Soyuz launch vehicle and commissioned in February 2021. The satellite (Norad ID 46492) has a polar orbit with a 526.6 km perigee, a 550 km apogee, an inclination of 97.8°, and a period of 95.3 min. The communication payload has 30 dBm output power capabilities at 868 MHz and 915 MHz. Lacuna Space is authorized to transmit over Europe, corresponding to a latitude ranging from 41° to 58°. The experiments were carried out in the city of Antibes, France (43.55, 7.13), from February to May 2021. Lacuna Space uses a Chirp Spread Spectrum modulation (LoRa) with a 250 kHz frequency bandwidth for the downlink. This waveform is robust to the Doppler frequency offset effect. A terminal using a trifilar antenna [32] and an electronic module based on the SX1262 transceiver was used as the terrestrial receiver. The receiving antenna provides a 2.5 dBic realized gain at zenith and was placed 10cm above the ground using a tripod. It should be noted that no additional radio frequency components (low-noise amplifier, filter, etc.) were used during the experiment, which confirms the capability to receive a broadcast signal from LEO orbit with a low-cost and low-power consumption terminal. The Received Signal Strength Indicator (RSSI) was extracted for every received packet. The left part of Figure 15 shows the RSSI versus elevation for 28 different passes. The same results are shown over a Europe map on the right of Figure 15, with every point being the satellite’s position when it transmitted the packet. The position of the terrestrial received is indicated with the gray marker. During the experiment between the satellite and the terrestrial receiver, a maximum communication distance of 1232 km was reached.

## 5. Discussion

The proposed structure was validated in space, and the non-deployable passive antenna was demonstrated to be able to cover multiple bands in the UHF frequency with a 3U CubeSat configuration. It should be noted that the antenna’s performance could be further enhanced using a lower-loss substrate. Due to the limited development time, we selected the FR-4 material to benefit from the space qualification and reduced production time. From a 3D electromagnetic solver simulation with a Rogers 4350b substrate (tan δ = 0.002), an average gain enhancement of about 1.5 dB is estimated. This study highlights the strong impact of the satellite structure on compact antenna performance. It was clearly shown that for both input impedance and gain, the CubeSat chassis modifies the distribution of the currents and thus changes the reflection coefficient and radiation pattern. The grounding of the proposed antenna with the CubeSat metal structure plays a crucial role in the antenna performance and should not be underestimated. The use of a characteristic mode analysis in the preliminary design phase of the structure would enable more comprehensive modeling and would reduce the integration risk. In comparison with the state of the art presented in Table 1, the proposed antenna offers a more compact form factor, which enables integration into a 1U cubesat face. The strong miniaturization of the antenna structure limits the operating frequency fractional bandwidth to 7.8%, which is sufficient for the targeted application. In the future, multi-band operations, including 400 MHz, L-band, and S-band, will also be considered.

## 6. Conclusions

A multi-band right-hand circularly polarized antenna integrated into a three-unit CubeSat has been designed, simulated, and measured. A dual substrate solution based on FR4 material has been manufactured to validate the simulation model. After integration into a 3U CubeSat structure, the antenna provides an RHCP configuration with aperture beamwidths of 134 °C and 144 °C at 870 MHz and 920 MHz, respectively. The proposed antenna has been successfully tested in flight with a communication range greater than 1200 km from an LEO CubeSat to a low-power terrestrial receiver.

## Figures and Tables

**Figure 1 sensors-23-05361-f001:**
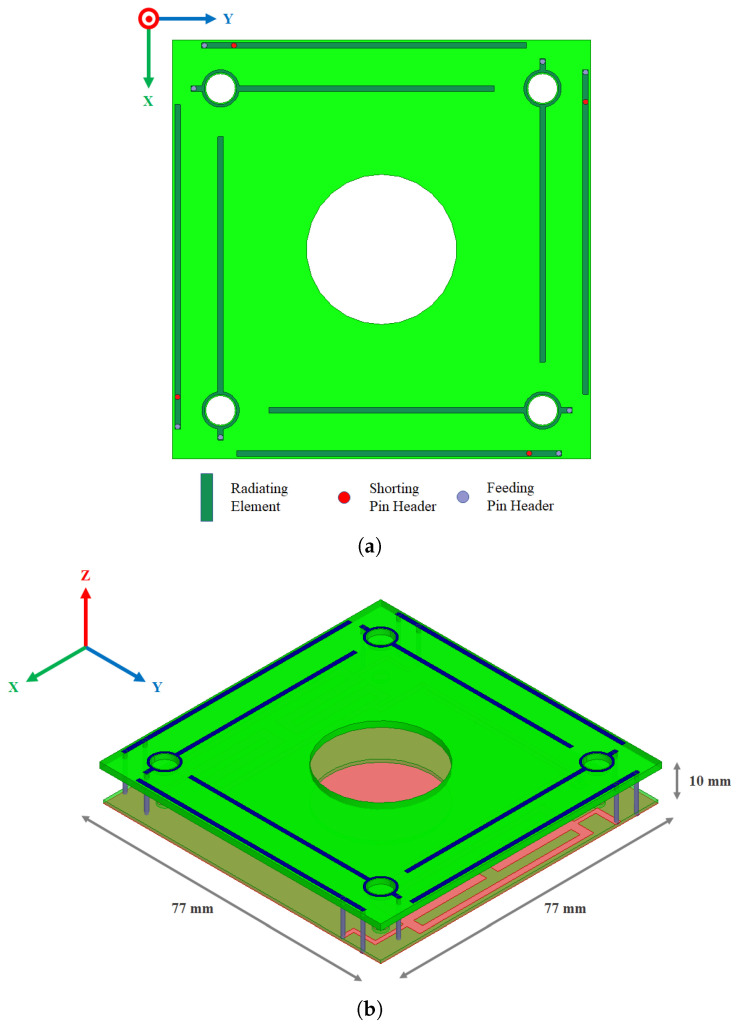
(**a**) Top view of antenna elements; (**b**) 3D view of the proposed antenna structure.

**Figure 2 sensors-23-05361-f002:**
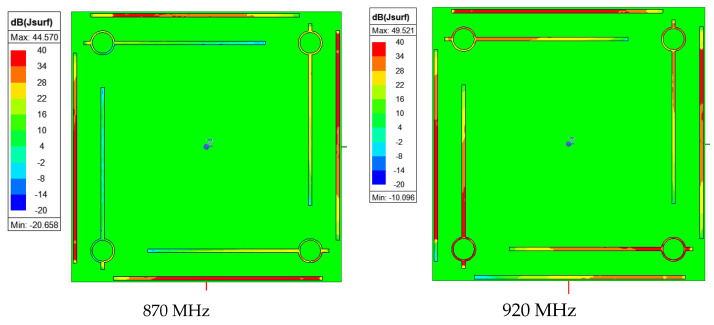
Surface current on the radiating element from the top view.

**Figure 3 sensors-23-05361-f003:**
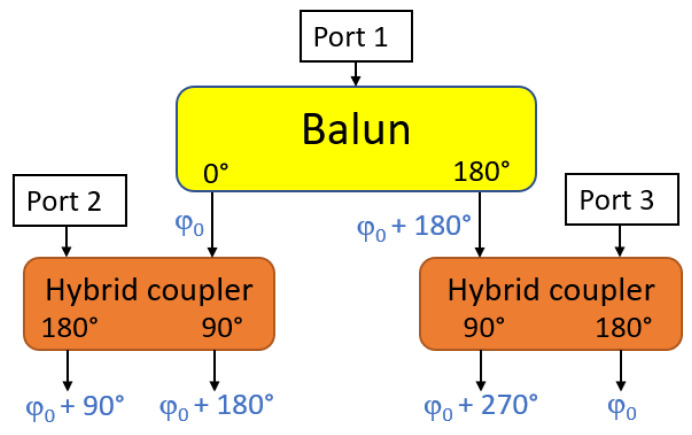
Block diagram of feeding network structure.

**Figure 4 sensors-23-05361-f004:**
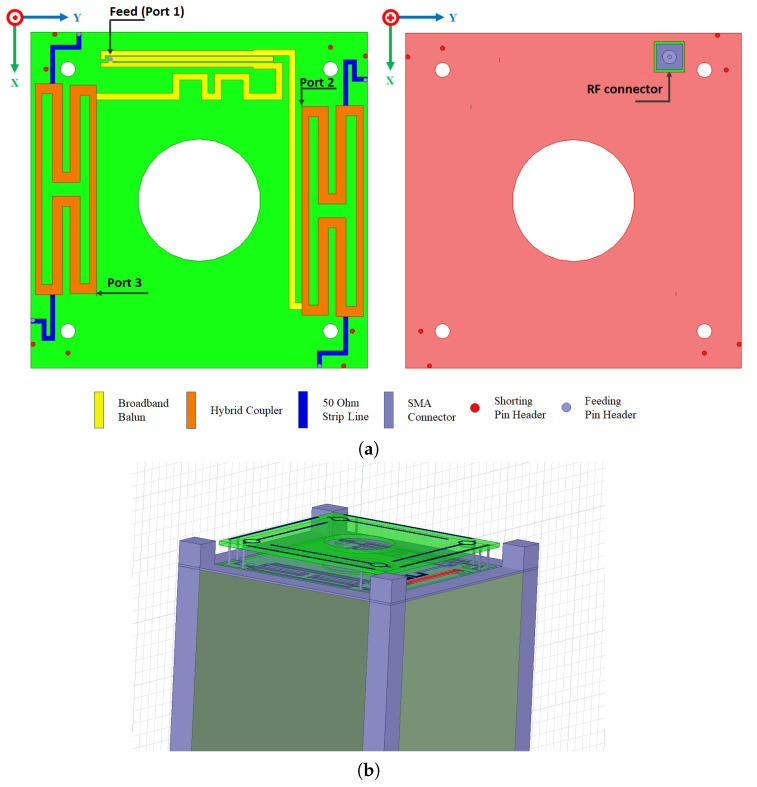
(**a**) Layout of the feeding circuit; (**b**) 3D view of the antenna model integrated within the 3U structure.

**Figure 5 sensors-23-05361-f005:**
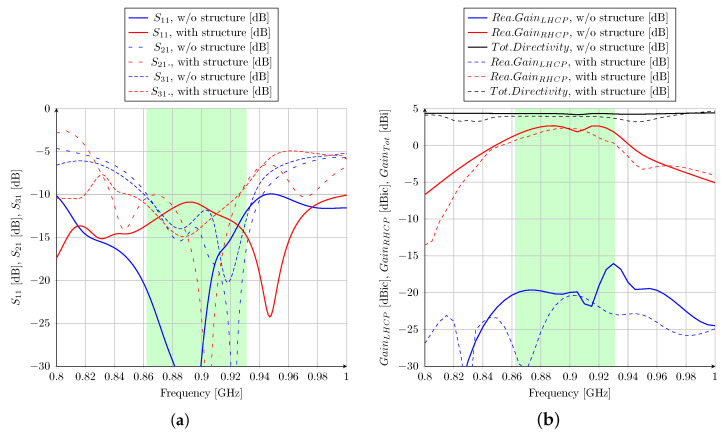
(**a**) S parameter simulation of the antenna with and without structure; (**b**) gain radiation simulation of the antenna with and without structure.

**Figure 6 sensors-23-05361-f006:**
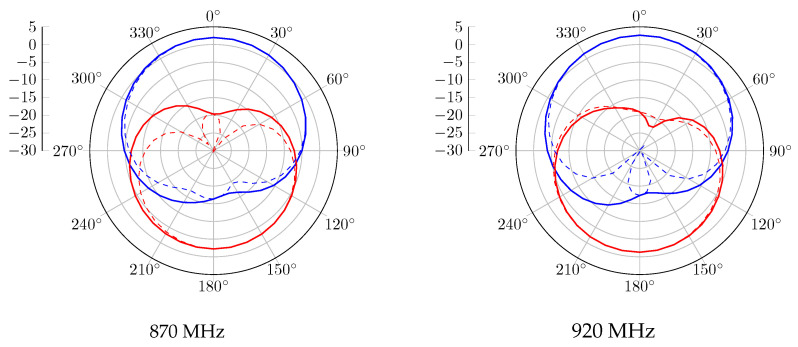
Gain radiation pattern simulation of the antenna without the structure, with RHCP in blue and LHCP in red, plain trace for phi = 0°, and dashed trace for phi = 90°.

**Figure 7 sensors-23-05361-f007:**
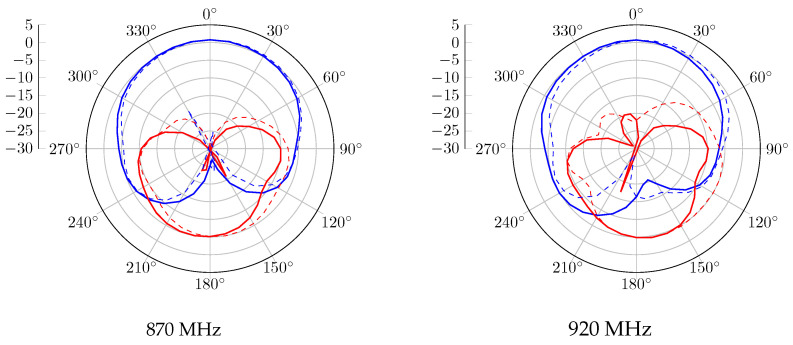
Gain radiation pattern simulation of the antenna with structure, with RHCP in blue and LHCP in red, plain trace for phi = 0°, and dashed trace for phi = 90°.

**Figure 8 sensors-23-05361-f008:**
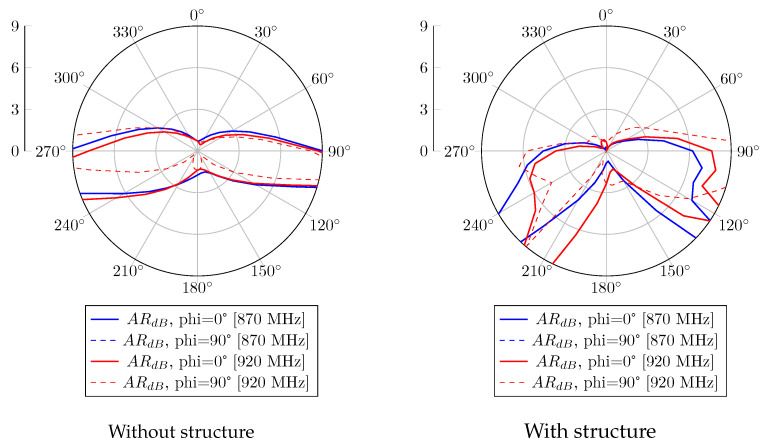
Axial ratio versus theta simulation of the antenna without and with structure.

**Figure 9 sensors-23-05361-f009:**
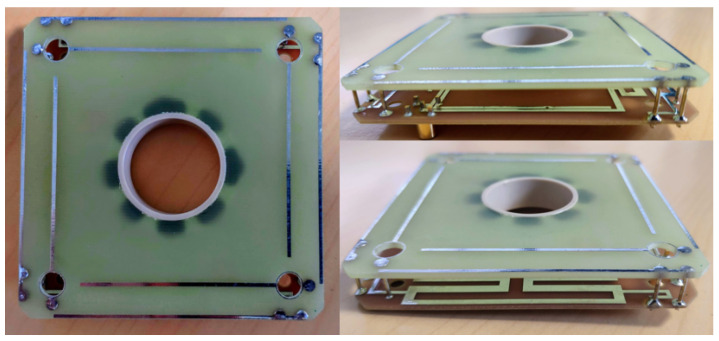
Picture of the prototype with: top view of the prototype; side view in hybrid coupler orientation; side view of the prototype in balun orientation.

**Figure 10 sensors-23-05361-f010:**
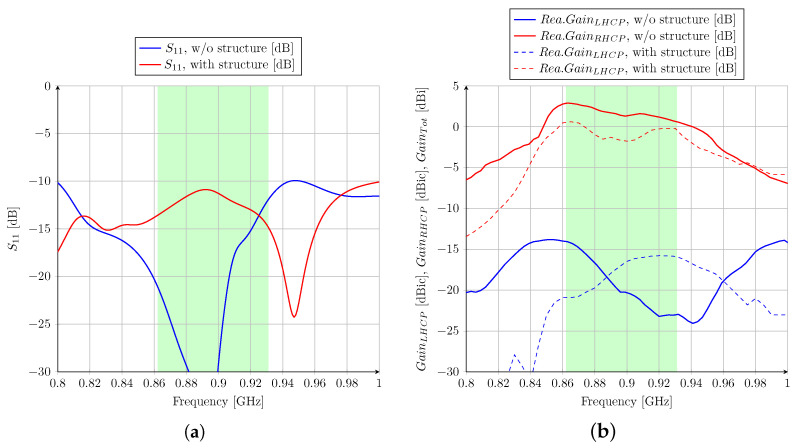
(**a**) S parameter measurement and (**b**) realized gain measurement of the antenna with and without structure.

**Figure 11 sensors-23-05361-f011:**
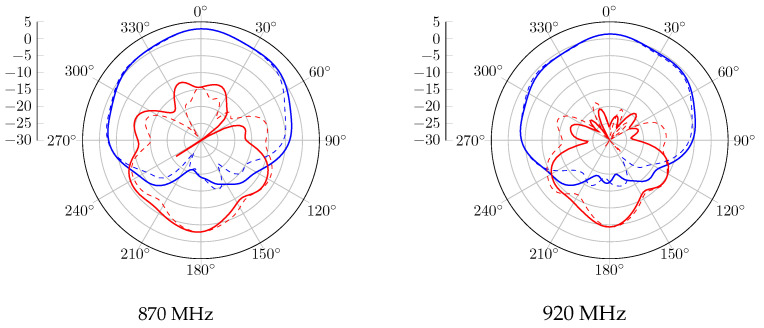
Measured gain radiation pattern of the antenna without structure, with RHCP in blue and LHCP in red, plain trace for phi = 0°, and dashed trace for phi = 90°.

**Figure 12 sensors-23-05361-f012:**
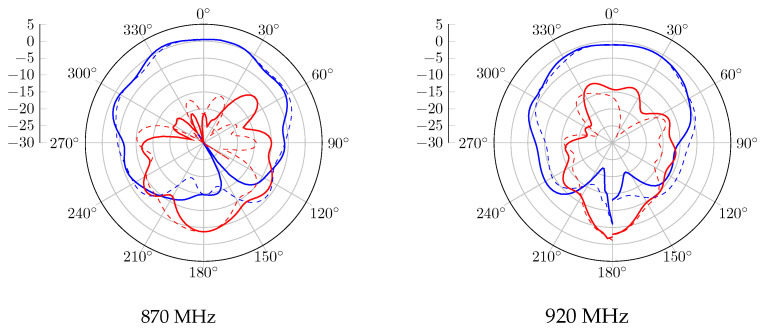
Measured gain radiation pattern of the antenna with structure, with RHCP in blue and LHCP in red, plain trace for phi = 0°, and dashed trace for phi = 90°.

**Figure 13 sensors-23-05361-f013:**
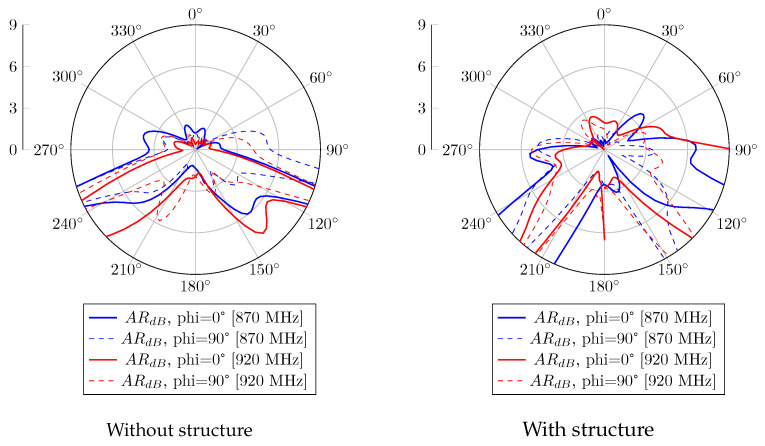
Axial ratio versus theta measurement of the antenna without and with structure.

**Figure 14 sensors-23-05361-f014:**
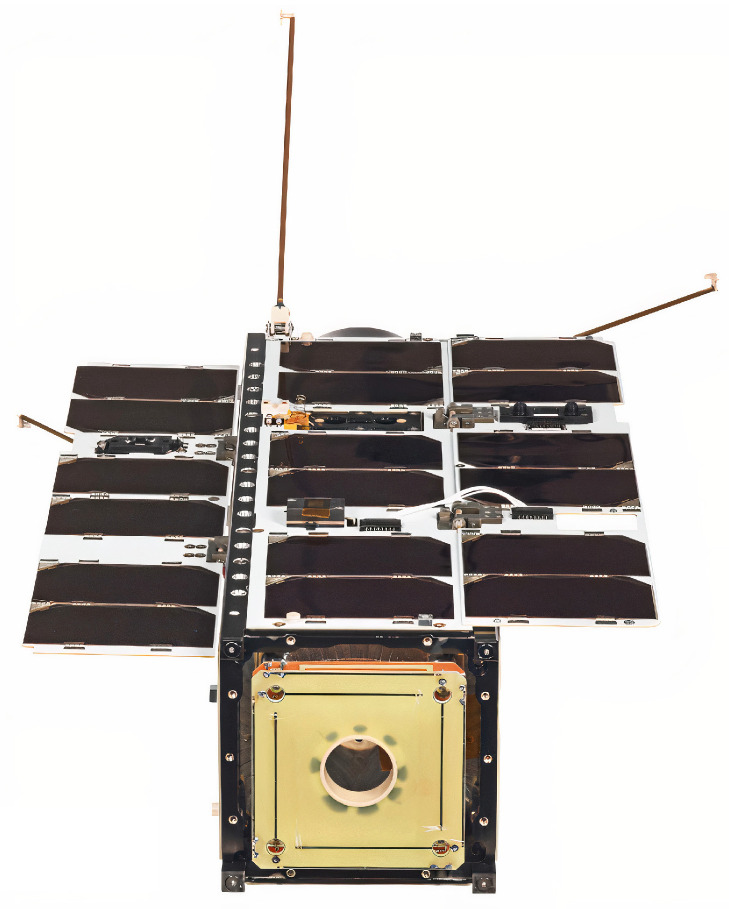
Final integration of the antenna within the 3U LS3 satellite with the proposed antenna on the front.

**Figure 15 sensors-23-05361-f015:**
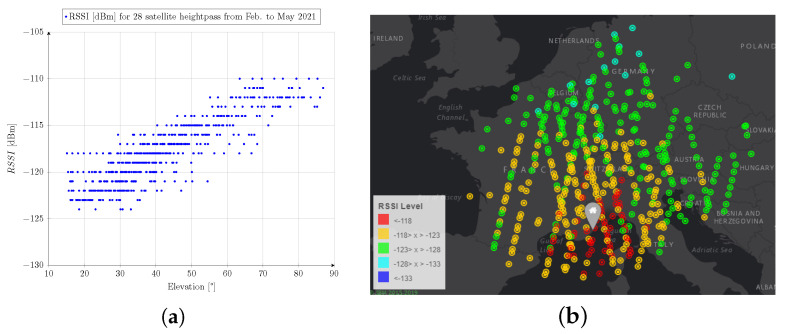
(**a**) Received strength indicator on the terrestrial node versus elevation. (**b**) Position of the satellite when LoRa packets are received in the Antibes area (gray marker).

**Table 1 sensors-23-05361-t001:** Comparison with the state of the art.

	[22]	[24]	[31]	This Work
**Center** **Frequency**	2.5 GHz	2 GHz and 8.5 GHz	850 MHz	900 MHz
**Dimension in λ**	0.83 × 0.83 × 0.06	0.4 × 0.4 × 0.055	0.47 × 0.47 × 0.5	0.22 × 0.22 × 0.03
**Reconfigurable feature**	No	No	Yes/Mechanical	No
**Impedance Bandwidth**	41%	12%	28.6%	7.82%
**3dB Axial Ratio Bandwidth**	41%	8%	N/A	7.82%

## Data Availability

Not applicable.

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
