# Peer review of "Compact UHF Circularly Polarized Multi-Band Quadrifilar Antenna for CubeSat"

_sensors, 2023, doi:10.3390/s23125361_

Round 1

Reviewer 1 Report

The authors present a detailed discussion on Compact UHF Circularly Polarized Multi-band quadrifilar antenna for CubeSat application. A prototype is fabricated and experimentally verified as well. This reviewer appreciates the quantum of experiments done to validate every data. However, the novelty of the structure of the antenna is not clear and its working mechanism is partially missing in the manuscript. The authros must add the near field analysis and the surface current distribution at different frequencies for the readers to understand the working mechanism of the antenna.

Minor editing is required in the English language.

Author Response

We want to thank the reviewer for the fruitful comments.

The surface currents are now provided to provide a better explanation of the working mechanism of the structure.

A comparison with the State of the Art shows the novelty of the structure.

Reviewer 2 Report

This manuscript presents a compact UHF CP antenna for CubeSat. The antenna performance was acceptable for real application with a integrated feeding network. Other comments:
1. The bandwidth of the antenna was good, while the gain bandwidth was a bit narrow.
2. The idea to achieve CP performance was a common method. The parasitic element effect can be analyzed with more details.
3. How about the AR vs theta?
4. The entire profile was a bit high, how to overcome this issue?

Author Response

We want to thank the reviewer for the fruitful comments.

We want to answer to the different reviewer concerns :

  1. The bandwidth of the antenna was good, while the gain bandwidth was a bit narrow.

The 3dB Peak gain bandwidth is similar to the matching bandwidth. This result has been clarified in the comments on the measurements.

2. The idea to achieve CP performance was a common method. The parasitic element effect can be analyzed with more details.

The surface currents are now provided to provide a better explanation of the working mechanism of the structure.

3. How about the AR vs theta?

We add 4 figures to show the AR vs theta results simulated and measured

4. The entire profile was a bit high, how to overcome this issue?

A comparison with the State of the Art shows the novelty of the structure.

The vertical profile appears to be very competitive compared with state of the art structure.

Reviewer 3 Report

The authors showed the circular polarization antenna for a CUBE-SAT. Overalls, this manuscript is well-structured and written. There are minor concerns as follows:

1. Please compare your antenna with state-of-art CubeSat antennas.

2. A 25 mm diameter ceramic ring is used in the proposed antenna. Did you include the ceramic ring in HFSS simulation? If not, please show the simulation results of the designed antenna with the ceramic ring.

3. How are Ports 2 and 3 of Figure 3 terminated with 50 ohms?

4. In my opinion, the simulation results of Fig. 8 are quite different from the measured results of Fig. 9. These differences should be addressed more well.

5. Please explain in more detail the following sentence:

"Lacuna Space is using a 250 kHz bandwidth LoRa modulation to avoid the Doppler frequency offset effect and an SF11 configuration (Spreading Factor - SF) to offer a good trade-off between sensitivity and time-on-air."

There are a few minor typos.

For example,

'dBiC' should be changed to 'dBic'.

TanD -> tan ?
30MHz -> 30 MHz

868MHz -> 868 MHz

Inspired -> inspired

KHz -> kHz

The Received -> the received, and so on.

Author Response

We want to thank the reviewer for the fruitful comments.

We will try to answer to the different reviewer concerns :

  1. Please compare your antenna with state-of-art CubeSat antennas.

A State of the art comparison was added in the discussion part to highlight the benefit of the proposed structure.

2. A 25 mm diameter ceramic ring is used in the proposed antenna. Did you include the ceramic ring in HFSS simulation? If not, please show the simulation results of the designed antenna with the ceramic ring.

We were able to validate in simulation that the ceramic ring has no influence in the electromagnetic antenna performance. A sentence was added to provide this information.

3. How are Ports 2 and 3 of Figure 3 terminated with 50 ohms?

Ports 2 and 3 are indeed matched to 50ohm with discrete resistors. A sentence has been added to provide the information.

4. In my opinion, the simulation results of Fig. 8 are quite different from the measured results of Fig. 9. These differences should be addressed more well.

Additional explanations about the difference between the two figures were added to the document to provide a more convicing explanation.

5. Please explain in more detail the following sentence:

"Lacuna Space is using a 250 kHz bandwidth LoRa modulation to avoid the Doppler frequency offset effect and an SF11 configuration (Spreading Factor - SF) to offer a good trade-off between sensitivity and time-on-air."

The sentence has been updated to clarify the modulation used in the downlink process.

The different typing errors have been corrected. Thanks again for the comments.

Reviewer 4 Report

This article presents a wide-band RHCP antenna design for the CubeSat. The antenna is quadrifilar with 4 Inverted-F elements and a sequential-rotation feeding. The structure can fit into a small size of 77x77x10 mm3, and the performance can meet the requirements of LoRa transmission covering 860-920 MHz in three continents. The real terrestrial-to-space communication link was measured in the city of Antibes, France from February to May, 2021, with reasonable performance. This article is valuable for field practitioners.

In addition, there are some minor suggestions for the authors to consider.

1.      The captions in the figure are invisible. They should be enlarged.

2. Double-check for possible typos or grammar errors. For example, there is a redundant S11 in lines 191 and 192 on page 5.

3.      The idea of the feeding circuit layout is said to be originated from [33], but the topology is still quite different. It is worth some effort to describe the layout design and especially, the simulated magnitude and phase at the four feeding points for the sequential rotation. 

Author Response

We would like to thank the reviewer for the fruitful comments.

We will try to answer reviewer comment step by step :

  1. The captions in the figure are invisible. They should be enlarged.

Thanks a lot, figures were enlarged

  1. Double-check for possible typos or grammar errors. For example, there is a redundant S11 in lines 191 and 192 on page 5.

Typo have been corrected

  1. The idea of the feeding circuit layout is said to be originated from [33], but the topology is still quite different. It is worth some effort to describe the layout design and especially, the simulated magnitude and phase at the four feeding points for the sequential rotation. 

Additional information on the circuit design has been provided to clarify the feeding circuit concept.

Reviewer 5 Report

The antenna is based on a quadrifilar structure, specifically designed for CubeSat applications. This design approach is likely a strength of the paper, demonstrating an innovative and potentially effective solution for satellite communication. Moreover the antenna is designed to cover the LoRa frequency bands of 868 MHz, 915 MHz, and 923 MHz, indicating a multi-band capability that is crucial for flexible operation in various countries.

I have no critical comments on the paper. The only suggestion is to place S parameters of the simulated and the designed antenna on the same plot to simplify the analysis.

Author Response

We would like to thank the reviewer for the fruitful comments.

Figures are been updated according to the different reviewer's comments in order to clarify the simulation and measurements.